# Developing the eMedical Student (eMS)—A Pilot Project Integrating Medical Students into the Tele-ICU during the COVID-19 Pandemic and beyond

**DOI:** 10.3390/healthcare9010073

**Published:** 2021-01-14

**Authors:** Joshua Ho, Philip Susser, Cindy Christian, Horace DeLisser, Michael J. Scott, Lynn A. Pauls, Ann M. Huffenberger, C. William Hanson, John M. Chandler, Lee A. Fleisher, Krzysztof Laudanski

**Affiliations:** 1Perelman School of Medicine, University of Pennsylvania, Philadelphia, PA 19104, USA; Joshua.Ho2@pennmedicine.upenn.edu (J.H.); Philip.Susser@pennmedicine.upenn.edu (P.S.); christian@email.chop.edu (C.C.); delisser@pennmedicine.upenn.edu (H.D.); 2Department of Anesthesiology and Critical Care, University of Pennsylvania, Philadelphia, PA 19104, USA; Michael.Scott@Pennmedicine.upenn.edu (M.J.S.); paulslynn@gmail.com (L.A.P.); William.Hanson@uphs.upenn.edu (C.W.H.III); lee.fleisher@uphs.upenn.edu (L.A.F.); 3Penn Medicine Center for Connected Care, Clinical Practices of the University of Pennsylvania, Philadelphia, PA 19104, USA; Ann.Huffenberger@pennmedicine.upenn.edu; 4Department of Neurology, University of Pennsylvania, Philadelphia, PA 19104, USA; john.chandler@uphs.upenn.edu; 5Leonard Davis Institute for Healthcare Economics, University of Pennsylvania, Philadelphia, PA 19104, USA

**Keywords:** COVID-19, critical care, medical education, telehealth

## Abstract

The COVID-19 pandemic has accelerated the demand for virtual healthcare delivery and highlighted the scarcity of telehealth medical student curricula, particularly tele-critical care. In partnership with the Penn E-lert program and the Department of Anesthesiology and Critical Care, the Perelman School of Medicine (PSOM) established a tele-ICU rotation to support the care of patients diagnosed with COVID-19 in the Intensive Care Unit (ICU). The four-week course had seven elements: (1) 60 h of clinical engagement; (2) multiple-choice pretest; (3) faculty-supervised, student-led case and topic presentations; (4) faculty-led debriefing sessions; (5) evidence-based-medicine discussion forum; (6) multiple-choice post-test; and (7) final reflection. Five third- and fourth-year medical students completed 300 h of supervised clinical engagement, following 16 patients over three weeks and documenting 70 clinical interventions. Knowledge of critical care and telehealth was demonstrated through improvement between pre-test and post-test scores. Professional development was demonstrated through post-course preceptor and learner feedback. This tele-ICU rotation allowed students to gain telemedicine exposure and participate in the care of COVID patients in a safe environment.

## 1. Introduction

Telemedicine and virtual care delivery expansion is inevitable, and the COVID-19 epidemic accelerated its implementation [1,2,3,4]. Current and future healthcare providers must be trained in virtual care delivery. Medical training, to date, has not emphasized competency in telemedicine [5,6]. In 2018, only one specialty, Child and Adolescent Psychiatry, included telehealth as part of its Accreditation Council for Graduate Medical Education (ACGME) Milestones [6]. Concurrently, 60% of medical schools surveyed by the Association of American Medical Colleges (AAMC) reported including telemedicine in their curricula that same year [6]. However, this number could be considered low as telemedicine is rapidly becoming a leading modality in healthcare delivery. It is also unclear how these educational programs are structured and assessed for efficiency.

The COVID-19 pandemic has been an unprecedented challenge to healthcare systems worldwide [1]. The demand for frontline providers able to deliver ICU-level care has exceeded supply, leading to the deployment of providers who lack sufficient intensivist training. These providers need support to deliver safe and effective care. Telemedicine has the potential to off-load and augment frontline providers, conserving time and medical equipment [4,6]. Tele-critical care, in particular, allows for a tiered system of healthcare delivery, where one intensivist or specialist is able to safely support multiple frontline providers, especially providers who may lack intensivist training but whose patients require ICU-level care [6].

Medical student involvement in direct patient care was suspended across many institutions early in the COVID-19 pandemic [7]. Virtual patient engagement, however, remained an option without risk of exposure while enabling student contributions to clinical care. Apart from the description of a “one-day clerkship” experience [8], formal programs or curricula targeting medical students during the pandemic have not been reported in the literature.

The COVID-19 pandemic demands both innovative methods for delivering care to critically ill patients as well as new training opportunities that provide for the safety of students [1,2,9]. In this report, we describe the development and initial outcomes of a tele-ICU elective course for post-clerkship medical students that emerged from a collaboration between students and faculty.

## 2. Materials and Methods

### 2.1. Development, Goals, Learning Objectives

The clinical elective, “eICU rotation: COVID-19 & Beyond”, was developed in March 2020 when in-person clinical activities for medical students were suspended for students at the Perelman School of Medicine (PSOM) because of the emerging COVID-19 pandemic. It was designed as a 4-week clinical rotation for post-clerkship students through a collaborative partnership among eICU faculty, curriculum leadership at PSOM and recruited 4th year students. The students took the lead in developing a curriculum that would enable medical students to remotely, safely and meaningfully participate in the care of critically ill COVID-19 patients. The goal of the elective was to enable students to acquire virtual patient care skills and understanding of critical care patient management through using a telemedicine platform and remote patient assessment protocols. Learning objectives for students were to: (i) assess and triage critically ill patients in a virtual setting; (ii) provide clinical care as a part of a remote interdisciplinary care; (iii) reflect on provider well-being in the midst of the pandemic; and (iv) describe the institutional challenges of providing care during a pandemic.

### 2.2. Clinical Setting

Students rotated remotely through a 20-bed medical-surgical ICU in the University of Pennsylvania Health System (UPHS) that at the time only provided care for critically ill patients with a confirmed COVID-19 diagnosis. The remote site was at a UPHS facility in Philadelphia 2.1 miles from the medical center and the medical-surgical ICU. ICU on-site staff included an intensivist physician, a critical care fellow, two advanced practice providers and a respiratory therapist. Tele-ICU staff at the remote location included a 24/7 available intensivist (eMD) and ICU nurse (eRN). Tele-ICU respiratory therapists (eRT), CA-3 anesthesiology residents (eRes) and medical students (eMS) were available on weekdays only. Tele-ICU staff routinely checked in on newly admitted ICU patients and could also be contacted directly for acute issues via an automated system (eCareManager; Koninklijke Philips N.V., Amsterdam, The Netherlands), direct telephone call or emergency alert button located in each ICU room. A camera, video screen and microphone were placed in every ICU room, allowing bidirectional visual and audio communication between on-site ICU staff and tele-ICU staff. Data from the camera, video screen and microphone were not recorded.

### 2.3. Rotation Clinical Activities and Responsibilities

Students (eMS) were assigned four 5-h shifts per week under the direct supervision of CA-3 anesthesiology residents with supervising faculty on site. There were two students per shift, each following one or two patients. During their clinical shifts, eMS reviewed the electronic health records (Epic Systems Corporation, Verona, WI, USA) of their patients using a checklist developed from University of Pennsylvania Health System clinical practice guidelines (Appendix A) and coded in the REDCap database [9,10]. Remote video examinations of critically ill patients were performed using eCareManager (Philips Healthcare, Amsterdam, The Netherlands) by the students. The system provides high fidelity audio and video examination and two-way communication capability. The high quality of the system ensures its effective application in clinical settings and is in use in our facility currently. After reviewing the patient chart and examining the patient with resident supervision, eMS presented cases to intensive care faculty physicians in the unit (on-site intensivists) or virtual tele-ICU physicians (eMD) via communication link or in person. Clinical interventions, defined as instances where students identified and proposed a change to the management of a patient, were recorded by students (Appendix A). These interventions were categorized as urgent if the intended response time was less than 10 min. Otherwise, they were categorized as routine. This classification is also used at the tele-ICU center. Students were also asked to document instances in which the faculty provided teaching on clinical decision-making pertinent to their patients. Social distancing of six feet and wearing surgical face masks were required at all times at the remote site.

### 2.4. Rotation Structured Educational Components

As students were engaged in remote but direct care of critically ill patients, they participated in weekly debriefing sessions over teleconference led by two emeritus faculty recruited by the PSOM administration. These debriefing sessions allowed students to reflect on patient care and professional and personal challenges encountered during the week. Students on-call at the time of the conferences were excused from clinical duties for the hour. In addition, students participated in small group sessions where they gave a case presentation and a topic presentation each week. These were held in-person at shift change in a large conference room with adequate social distancing. Students were also instructed to post one journal article that had impacted their patient care management plans on an online discussion board and comment on others’ posts. Finally, students were assigned a reflection essay as a capstone assignment for the experience. The reflection essay was intended to summarize key teaching points from the rotation and notable clinical cases. It was also designed for students to reflect on their professional development throughout the rotation.

### 2.5. Rotation Assessment

Both students and preceptors were asked to evaluate the rotation using anonymous REDCap-based surveys containing both multiple choice seven-point Likert scale questions and open-ended questions (Appendix B and Appendix C). In addition, students completed both pre-rotation and post-rotation electronic assessments designed by the faculty to assess medical knowledge in the critical care setting.

### 2.6. Ethics Statement

This study was approved by the Institutional Review Board of the University of Pennsylvania.

## 3. Results

### 3.1. Overall Experience

The logistics of starting a clinical rotation and educational course during the early surge response delayed the course by one week, leading to a three-week pilot rotation. Five students took the rotation in April 2020. During the three weeks of clinical shifts, the students followed 16 patients, generating 73 notes and 70 contributions to patient management.

### 3.2. Patient Demographics

The patient clinical characteristics are presented in Table 1. Of the 16 patients followed by the students, most were above the age of 60 (88%), male (56%) and black (88%), per chart record.

### 3.3. Clinical Interventions

Students initiated a total of 70 clinical interventions, the majority of which were routine (95.6%) rather than urgent. The breakdown of interventions is detailed in Figure 1A,B. The most common clinical interventions were ventilator adjustments (17 instances, e.g., switching to low-stretch protocols) and medication adjustments (14 instances, e.g., adding insulin, adjusting pressors). eMS were able to ensure adherence to protocols and best practices for a variety of clinical concerns using the checklist (Appendix A).

### 3.4. Student Assessment: Professional Development

Four of the five students evaluated the rotation one week after the rotation ended (Appendix A). Students reported that the course improved their ability to manage ICU patients and evaluate COVID-19 patients and that the course had a high educational impact. All of the responding students had positive responses to the open-ended questions regarding the debriefing sessions, EBM assignments and interactions with eMDs and residents. The rotation empowered the eMS, demonstrated by high agreement with the statementsuch as “The course improved my ability to identify critical patients”. All respondents agreed with the statement “My well-being was a priority for the course administration during this course”. All respondents strongly agreed with the statements “The course improved my ability to work effectively in a digital health care delivery setting (telemedicine)” and “My ability to work with virtual (e-) providers has increased”. No students reported issues with technical aspects of the system (poor audio or video).

### 3.5. Student Assessment: Medical Knowledge

All eMS reported feeling capable of evaluating and managing ICU and COVID patients after this rotation. The assessment of their knowledge before and after the rotation showed improvement (Knowledge_baseline_ = 15.8 + 2.2 vs. Knowledge_post_ = 19.8 + 0.3; t[−3.2] = 00; *p* = 0.023). This assessment was corroborated by subjective agreement in the survey questions “I was able to engage inpatient care at a level consistent with my capabilities” and “this course improved my ability to assess and manage ICU patients”.

### 3.6. Preceptor Assessment of the Rotation

The provider post-rotation survey revealed a heterogenous perception of eMS by attendings, residents and fellows. Although both attendings and residents found interactions with eMS valuable, fellows, functioning in-unit in a role similar to that of attendings, were more heterogeneous in their responses. An open-ended prompt “what other feedback would you like to provide about this medical student rotation?” offered insight as the fellows, the group with the lowest scores, explained that they had not been made aware of the initiative or of the expectations on themselves or the students. One fellow stated that, when they did have “time to chat” with the student, they found the eMS’ “clinical reasoning, professionalism, communication skills, and thoughtfulness to be impressive” and suggested that “improving the preparation of preceptors would enhance the experience and interest in participation for preceptors”. The rotation was perceived as innovative by almost all respondents. One attending even remarked that the rotation was “much needed” and “forward-thinking”.

## 4. Discussion

This pilot rotation sought to provide a way for students to contribute to direct COVID-19 patient care remotely under supervision while building valuable critical care and telehealth skills. To the best of our knowledge, this is the first report describing a medical student tele-ICU rotation during the COVID-19 pandemic. Given the rise of telemedicine during the pandemic and the growth in telemedicine’s medical student involvement, it is essential to evaluate the educational value of this type of student experience in the critical care setting [6,8]. This clinical rotation was a safe and innovative method of delivering clinical training to medical students during the COVID-19 pandemic. Students could engage in the care of COVID patients and participate in clinical management without any exposure to the virus or the need for additional PPE. Over the three-week rotation, medical students provided 70 contributions to patient management, most commonly medication and ventilator adjustments. Providers viewed the course as innovative, while in-unit fellows were less favorable, not knowing about the course prior to its initiation. This underscores the importance of stakeholders understanding the goals and objectives of an innovative course before its implementation. The medical students noted increased confidence in their ability to manage critically ill patients and work with telehealth technology while feeling like their wellness was a priority throughout the course. This course illustrated the value of a telehealth critical care rotation and commitment to medical student education in the height of the pandemic.

There were several limitations to the pilot rotation. The course was only able to run once with five medical students as the COVID-19 surge resolved earlier than expected and the UPHS healthcare system and the medical school moved forward with post-surge plans. Despite the paucity of participants, the study is a demonstrator of the benefits of medical students participating in a similar program. Medical systems can benefit from utilizing them to reduce the operational backlog. Students gain important expertise in tele-medicine, an essential part of future healthcare. Regarding the quality of data and analysis, several nuances of the data were not communicated clearly to survey respondents. For example, for documented interventions, the proportion of “urgent” interventions is likely underestimated. The routine and urgent variables were not adequately defined prior to the study’s initiation, with respondents defaulting to “routine”. Also, while the survey was intended to capture cases of students receiving teaching from preceptors, this intent was not adequately conveyed, and teaching data is limited. Furthermore, several indirect benefits of the rotation can only be estimated. For example, interventions such as bundling lab draws were not captured by the checklist. Lab bundling limits patient contact and PPE use, increasing efficiency and reducing potential virus transmission.

If further surges arise in the future, it is likely that medical students will again be called upon to assist with the care of COVID-19 patients. Alternatively, the rotation can be modified to run outside of the COVID-19 context. One of the tools to provide the student with telemedicine interaction is virtual or augmented reality. These technologies allow for the simulation of almost any healthcare environment. Additionally, virtual and augmented existence may demonstrate conditions where the telemedicine system underperforms providers’ increasing resilience. The need for telemedicine in the delivery of care and UME training in telemedicine will remain long term. Future patients and care settings may also require flexibility, and this initiative can be adapted for those settings. We look forward to iterating our rotation, based on our initial experiences as well as emerging telemedicine literature. There is an unmet need to educate future providers about effective telemedicine utilization.

Outstanding questions include the portability of the training based on one digital platform to other software platforms and how this form of clinical engagement affects professional identity development. Due to the virtual nature of the interaction, students could not physically examine or talk to their critically ill and intubated patients. This new type of provider-patient relationship requires further investigation. While we provided support via unstructured debriefing sessions and informal “check-ins”, virtual provider processing and grief must be explored to ensure eMS are adequately supported in their role.

## Figures and Tables

**Figure 1 healthcare-09-00073-f001:**
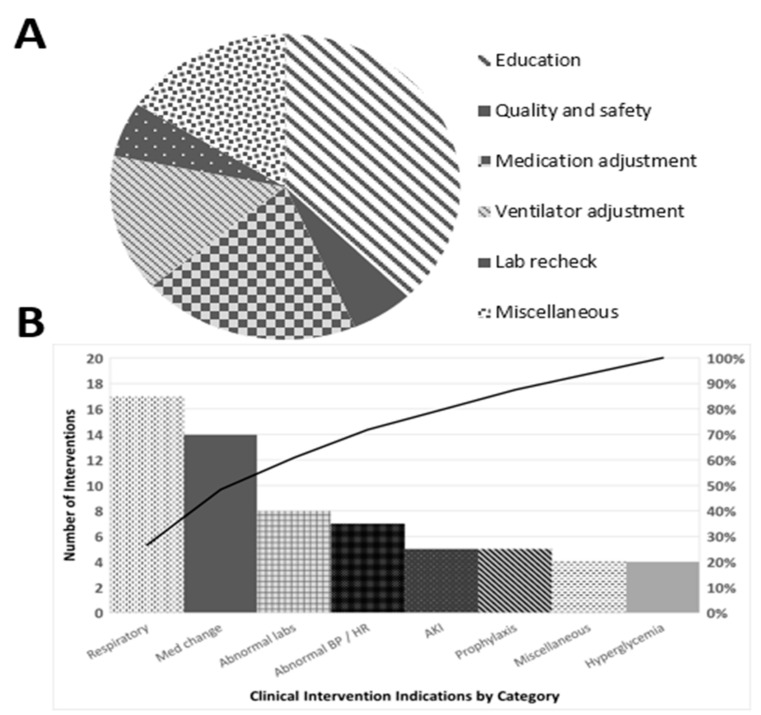
Interventions recorded by eMS: (**A**) distribution of category of interventions; and (**B**) Pareto chart of the indications for clinical interventions.

**Table 1 healthcare-09-00073-t001:** Demographics of the cohort of patients assigned to the medical students in a tele-ICU rotation, Perelman school of Medicine, April 2020.

Demographics	Patients (*n*)	Patients (%)
Age		
40–49	1	6%
50–59	1	6%
60–69	7	44%
70–79	4	25%
≥80	3	19%
Gender		
Female	7	44%
Male	9	56%
Race		
White	2	12.5%
Black	14	87.5%
Survivorship		
Yes	11	69%
No	5	31%
Discharged		
Yes	5	31%
No	6	38%
Deceased	5	31%
Length of Stay		
0–9 days	2	13%
10–19 days	7	44%
20–29 days	5	31%
≥30 days	2	13%

## Data Availability

The data presented in this study are available on request from the corresponding author. The data are not publicly available due to details of IRB consent.

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
