# Peer review of "Developing the eMedical Student (eMS)—A Pilot Project Integrating Medical Students into the Tele-ICU during the COVID-19 Pandemic and beyond"

_healthcare, 2021, doi:10.3390/healthcare9010073_

Round 1

Reviewer 1 Report

Dear Authors,

Thank you for allowing me to revise your innovative manuscript entitled: Developing the eMedical Student (eMS) – integrating medical students into the tele-ICU during the COVID-19 pandemic and beyond.

The problem with medical education during the COVID pandemic is a serious issue, and I compliment the authors on having highlighted the issue with this original manuscript. However, I see some drawbacks to this education model since the system is informative, but I don't see the needed potentials for train day after day for a young fellow. Please add some word about the need for a more innovative and realistic simulation environment based on Augmented Reality or Virtual Reality. The student's level of interaction and participation represents an unmet educational need to train the next generation of physicians.

Regarding the data, I suggest highlighting the title and the conclusions that this represents a pilot experience and that 5 students are not sufficient to state anything more than the need to design new educational pathways that encompass telemedicine and new technologies. Despite this important limitation, however, the manuscript envisions changes in educational methods that should engage scientific societies and academies to be really effective; please add a short sentence about it.

My best regards

Author Response

We address the issue of virtual reality as a way to augment the experience. The limitations related to the small sample have been highlighted in the title and discussion. We also added two sentences about implementing virtual reality as the way to augment the tele-ICU experience

Reviewer 2 Report

Dear authors,

I read your article with interest.
Because of covid-19, I think the need for telemedicine is increasing worldwide.
I provid you a few comments to help improve your article.

I think citation number usually does not appear in the abstract.

Line 37
It is better to spell out ACGME and AAMC for non-medical readers to understand.

Line 37
>Concurrently, 60% of medical 37 schools surveyed by the AAMC reported including telemedicine in their curricula that same year....
I wonder how the authors perceive "60%". Perhaps the authors feel it low.

I would like a little more detailed explanation of the tele-critical care system environment.
For example, can the sound be heard clearly or can the image be seen clearly? Isn't the operation complicated?
If possible, illustrate the tele-critical care system used in this study in a diagram. That is easier for the reader to imagine. In the future, when someone would like to adopt the method of this study, the figure will help reproduce the method.

Line130
age of 60 (88%), male (56%), and black, per chart record (88%).

age of 60 (88%), male (56%), and black (88)%, per chart record.
correct?

The letters in the figure might be too small to read.

Author Response

  1. The abstract was screened for citation
  2. ACGME and AAMC were spelled out
  3. We addressed the issue of telemedicine growth in the introduction.
  4. We decided against incorporating figures about system performance. We believe that this may suggest that the proprietary system used in our study is best. We want to avoid the bias or this suggestion as we believe that our findings is universal. Instead, we clarify several statements in the text to better describe the system.
  5. Figure was enlarged